# Perception of the Body Image in Women after Childbirth and the Specific Determinants of Their Eating Behavior: Cross-Sectional Study (Silesia, Poland)

**DOI:** 10.3390/ijerph191610137

**Published:** 2022-08-16

**Authors:** Mateusz Grajek, Karolina Krupa-Kotara, Martina Grot, Maria Kujawińska, Paulina Helisz, Weronika Gwioździk, Agnieszka Białek-Dratwa, Wiktoria Staśkiewicz, Joanna Kobza

**Affiliations:** 1Department of Public Health, Department of Public Health Policy, Faculty of Health Sciences in Bytom, Medical University of Silesia in Katowice, 41-902 Bytom, Poland; 2Department of Epidemiology, Faculty of Health Sciences in Bytom, Medical University of Silesia in Katowice, 41-902 Bytom, Poland; 3Department of Human Nutrition, Department of Dietetics, Faculty of Health Sciences in Bytom, Medical University of Silesia in Katowice, 41-808 Zabrze, Poland; 4Department of Technology and Food Quality Evaluation, Department of Dietetics, Faculty of Health Sciences in Bytom, Medical University of Silesia in Katowice, 41-808 Zabrze, Poland

**Keywords:** food restriction, body shape, body image, women, pregnancy, eating disorders, eating behaviors, TFEQ-13

## Abstract

Background: Episodes of loss of control over eating during pregnancy affect up to 36% of women during this period. Many women experience natural concerns about weight gain and body-shape changes during pregnancy and the postpartum period, and food cravings and fluctuations in eating patterns during these periods are physiological phenomena. However, pregnancy and the postpartum period may be an additional determinant of eating disorders. Women who perceive their own bodies as significantly deviating from the presented ideal are more likely to experience anxiety/anxiety related to it and also tend to exhibit abnormal eating behaviors. The perception of one’s body figure also plays an important role in maintaining psychological balance Aim: The aim of this study was to assess the perception of body image by postpartum women. The essence of the study was to see if there were psychomarkers associated with lack of control over eating, food restriction, and emotionally motivated eating in the study population. Material and methods: The study was conducted during the fall and winter of 2021. A total of 288 women participated in the study. The age of the subjects ranged from 21 to 45 years. Results: It was found that 198 women (68.8%) were dissatisfied with their current body weight and figure. Respondents with higher post-pregnancy body mass index showed dissatisfaction with their body shape (49.8%; H = 13.042; *p* = 0.001). Both body satisfaction and BMI level were significant components of the occurrence of pathological phenomena associated with the TFEQ-13 subscales (*p* < 0.05). Conclusions: Excessive focus on food restriction as well as lack of control over eating had some association with negative self-perception of body image, mainly in the form of body weight dissatisfaction. Eating behaviors showed an association with BMI level and weight satisfaction after pregnancy.

## 1. Background

During pregnancy, childbirth, and the postpartum period, many pathophysiological processes can occur in the body of every woman. The birth of a child brings with it changes in various spheres of family, professional, social, and economic life. Both pregnancy, childbirth, and puerperium are physiological states that have great influence on the quality of life of women [1]. It is important to analyze the factors negatively affecting a woman’s life during this period and especially after childbirth [2]. The physiological changes that accompany pregnancy favor, among other things, an increase in metabolism and energy demand [1,2]. As a result of the intensive action of hormones in pregnant women, the distribution of fat tissue and the increase in body weight, among other things, undergo significant modifications [1,3]. These activities are aimed at adapting the mother’s organism to the needs of the developing fetus and a smooth labor process. The variety of physiological changes that occur during pregnancy has an enormous impact on the subjective assessment of the mother’s quality of life [1,4].

In the pre- and perinatal period, predisposing factors for the development of poor eating habits are multiple pregnancy, biological age of parents (35–39 years), planning offspring, perinatal complications, delivery between 22–37 weeks of gestation, cesarean method, genitourinary tract infection (before pregnancy), gastrointestinal, metabolic-endocrine diseases, body mass index in women below 18.5 kg/m^2^ and above 25 kg/m^2^, as well as psychosocial factors concerning interpersonal relations, degree of satisfaction with motherhood, and perception of own body [5,6,7,8,9]. The changes associated with negative body image perception, as well as dysmorphophobia in specific clinical cases, have a psychological basis leading to eating disorders. They are conditioned by stress, feelings of anxiety, low socioeconomic status, depressive and dissociative states, and addictions. This also contributes to elevated levels of destructive tendencies in the relationship with food in the form of loss of self-control in the pre- and post-pregnancy periods [10]. The pre-conceptional, perinatal period including the first and second trimesters of pregnancy and the postpartum period among women is characterized by both loss and excessive control over food intake and analysis of food quality as well as consumption of food products without a sense of self and guided by emotional action, considered to be independent of thought processes. Consequently, these things shape an obsessive degree of monitoring behavior in the context of women’s diet [11]. One of the factors for postnatal depression is past or current eating disorders. This is a group of mental health disorders characterized by severe eating behavior disorders that significantly impair health and psychosocial functioning [12]. The prevalence of eating disorders in pregnant women ranges from 0.6% to 4.5% [13]. These disorders include, for example, anorexia and bulimia. These disorders often occur in women after childbirth because concerns about an inadequate figure are reinforced by the physical and psychological changes that occur during pregnancy [12]. These physiological changes associated with pregnancy can consequently cause an unhealthy obsession with weight loss and lead to postpartum depression [13]. Given their destructive effects on women and their infants, early identification of eating disorders and appropriate perinatal care is very important for women’s mental health during the prenatal and postnatal periods [12]. 

Not for every woman is the period of pregnancy a positive experience, and not every woman views the changes in her body through the prism of the developing fetus but mainly through the prism of weight gain and becoming an “unattractive” woman. It would seem that putting one’s own body image above proper nutrition for the health of the baby is rather unprecedented. Pregnancy represents such a period in a woman’s life when weight gain is even expected. However, women do not always fully accept the changes that occur in their body image. The results of some studies suggest that satisfaction with one’s body during pregnancy is reduced in some women. According to Downs et al., many pregnant women experience ambivalent feelings about their bodies, including worry due to higher body weight. Both social observations and scientific research have drawn attention to the phenomenon of pregorexia. Pregorexia (English: pregnancy, Latin: anorexia) is a relatively recent term used to denote anorexia during pregnancy and just after childbirth; it involves feelings of guilt caused by weight gain. Women affected by this disorder make efforts at all costs to look thin during pregnancy and postpartum. In order to keep their body weight as low as possible, they use hunger strikes, drastically reduce the daily caloric supply of the food they eat, force vomiting, and undertake intense exercise. It is believed that pregnancy is not the cause of the problem but only an activating stimulus. To date, the impact of physical changes during pregnancy on pregnant women’s perceptions of their bodies has yet to be clearly established. Directions of the research undertaken show that under the issue of body image in pregnancy lies not only satisfaction or dissatisfaction with the physical changes that occur during pregnancy but also the dimension of autonomy–symbiosis with the unborn child and the relationship between the body and the mental representation of the child, partner, and parent. On the issue of satisfaction with body image, researchers obtain contradictory results: on the one hand, some studies suggest higher bodily self-esteem during pregnancy, acceptance of the shape of one’s body, and physical changes associated with pregnancy. On the other hand, some women may experience anxiety, dissatisfaction with weight gain, and worry; this is most often the case for women whose occupation is related to physical appearance. The danger arises when a woman decides to lose weight during pregnancy and exercise too much, which can cause pregnancy complications and negatively affect the health of the baby [14,15,16,17,18,19].

Loss of control over eating (or, conversely, increased control over eating), feelings of powerlessness, and inability to control the amount or type of food consumed are among the most commonly reported eating behavior disorders. These episodes are often associated with weight gain, obesity, and psychological stress. Therefore, loss of control or increased control over eating is often considered the pathology underlying excessive eating [20]. Evidence suggests that perinatal health outcomes are compromised in women with a history of eating disorders [21,22]. Episodes of loss of control over eating are common in the general population, with estimates suggesting that up to 36% of women experience them during pregnancy [23]. Accompanying dysfunctional behaviors such as highly restrictive eating, episodes of overeating, excessive exercise, or laxative use depend on the subtype of eating disorder and often upset the body’s homeostatic balance [24]. Many women experience concerns about weight gain and figure change during pregnancy and the postpartum period, and food cravings and fluctuations in eating patterns during these periods are physiological. It is apparent, therefore, that pregnancy and the postpartum period present additional challenges for women with eating disorders [25,26].

Given the above, the study aimed to analyze postpartum women’s perception of their body image. The essence of the study was to find out whether psycho-markers related to excessive weight control or lack of them are present in the study population and whether there are relationships between them.

Based on the set main goal, the following specific goals can be formulated:1.Determinants of the BMI value with possible food restriction, uncontrolled eating, or emotional eating behavior in the studied population;2.Dissatisfaction with the body image after pregnancy and nutritional restrictions, lack of control over eating, or emotionally motivated eating behavior in the study population;3.Type of delivery and eating-restriction behaviors, lack of control over eating, or eating motivated by emotions in the study population;4.The method of a child’s stone after birth and food restriction, lack of control over eating, or emotional eating behavior in the study population.

The following working hypotheses were used in the study design:(A)Eating restriction depends on one’s BMI, dissatisfaction with one’s body, type of delivery, and the way the baby is fed.(B)Lack of control over eating depends on BMI, dissatisfaction with body appearance, type of delivery, and the way the baby is fed.(C)Eating under the influence of emotions depends on BMI, dissatisfaction with the appearance of one’s own body, the type of delivery, and the way the child is fed.

## 2. Material and Methods

### 2.1. Inclusion Criteria

Originally, 306 women participated in the study. The surveyed women were informed about the purpose of surveying them and the nature of the research. The participants voluntarily agreed to participate in the study, and therefore, they could also withdraw from the study at any time without giving any reason and without any consequences. The criterion for inclusion in the study was the birth of a child no earlier than 12 months before participation in the study and mental or somatic illnesses that could affect your perception of one’s body or its real size. The final analysis was based on the responses of 288 women. Based on the sampling calculation, the selected population was estimated to be representative assuming an significance level of α = 0.95%, fraction of 0.5, and a maximum error of 0.03.

In view of the ongoing COVID-19 pandemic, the study was conducted using a proprietary questionnaire, which was distributed by computer-assisted web interview (CAWI) to pregnant women in parenting groups via Facebook. At the beginning of the survey, women were given information regarding the purpose of the study and also on how to complete the questionnaire. The survey was conducted from September 2021 to the end of December 2021. All questionnaires were completed correctly.

The main criterion for inclusion in the study was the woman’s age ≤ 18 years. Women who were at least in the second trimester of pregnancy and up to one year after delivery were included in the study, as it is assumed that a period of 12 months is generally the time to return to the pre-pregnancy figure.

Participation in the study was anonymous and completely voluntary, based on so-called “spontaneous reporting”. The study complies with the provisions of the Declaration of Helsinki. The design of the study in light of the Act of 5 December 1996 on the professions of physician and dentist (Journal of Laws of 2011 No. 277, item 1634, as amended) is not a medical experiment and does not require the approval of the local bioethics committee. Approval from the Bioethics Committee of the Silesian Medical University in Katowice (ID. PCN/CBN/0052/KB/127/22) was obtained for participation in the study.

### 2.2. Research Tool

The research tool consisted of three parts: the first containing a metric, where sociodemographic and anthropometric data were collected; the second where data on current health status, the number of children born, the time and type of birth and how the child was fed, and satisfaction with one’s own body (body weight) were collected. Body satisfaction was assessed by subjective ratings of body satisfaction and body weight. Respondents expressed their opinion as to whether they were satisfied with their current body shape and weight, what they would like to change and why, and whether they planned to return to their pre-pregnancy body weight. The third part consisted on the TFEQ-13 questionnaire.

The TFEQ-13 questionnaire was used as a tool for analyzing the relationship with food. It consists of 13 questions included in 3 subscales: 5 questions concern food restriction, then 5 questions concern lack of control over eating, and 3 questions are directly related to emotional eating. The questionnaire contains standardized answers on a 4-point unidimensional scale from 0 to 3. The respondent marks the most defining statement next to each question according to the Likert scale assumptions: “definitely yes”, “rather yes”, “rather no”, and “definitely no”. The values are calculated separately for each subscale; the interpretation in terms of the examined psycho-marker can be seen in the table below (Table 1). The higher the score obtained, the greater the intensity of restriction eating and the lack of control over eating or eating under the influence of emotions [27].

### 2.3. Test Procedure

The study was conducted in three stages. The first stage was a pilot study, during which 30 randomly selected women were asked to complete a questionnaire to check whether all questions were understandable. The majority of questions were found to be clear and understandable by the respondents, while questions that were indicated by at least 2 respondents as not understandable or unclear were removed or framed. Stage two was questionnaire validation by distributing the questionnaires twice to a randomly selected group of 30 women. An interval of 2 weeks was maintained between the collection of the questionnaires. The responses to the same questions were checked for consistency. To assess the reproducibility of the results obtained with the used questionnaire, the value of the parameter ϰ (Kappa) was calculated for each question in the questionnaire—for 61.3% of the questions, a very good (ϰ  ≥  0.80) concordance of answers was obtained, while for 31.7% of the questions, a good (0.79 ≥ ϰ  ≥  0.60) concordance of methods was obtained. The final stage of the study was to conduct the actual test (general survey took place one month after the piloting to avoid the freshness effect).

For the test conducted, reliability of 0.88 expressed by the α-Cronbach coefficient was obtained. The power of the test was estimated to be 0.812. The power of the test was based on the key elements of the study: sample size, effect size, and significance α-level. It was assessed that the result obtained indicates high power of the test, which increases the precision and reliability of the estimates.

Due to the ongoing COVID-19 pandemic, the study was distributed by CAWI (computer-assisted web interview) method among women in parenting groups through the social networking site Facebook. In order to avoid the phenomenon of bot/random responders, the necessity of giving answers to all questions was applied; moreover, different types of questions (closed, open, semi-open) were used. In important aspects, the method of cross-questioning was used, which excluded the possible preparation of data by bots. In addition, the survey was available in thematic Facebook forums to which only users approved by the administrator had access; finally, after completing the survey, it was necessary to provide an alphanumeric or image code CAPCHTA.

The general survey took place one month after the piloting to avoid the freshness effect.

### 2.4. Statistical Analyses

All calculations were performed using Statistica 13.3 software (TIBCO Software Inc., Palo Alto, CA, USA). The Shapiro–Wilk test was used to determine the normality of the distribution of characteristics in the population. The test did not support the hypothesis of normality of distribution. The Kruskal–Wallis test was used for further analyses. Statistical significance was determined at *p* = 0.05.

## 3. Results

### 3.1. Study Group

The study was conducted during the autumn and winter of 2021. A total of 288 women were included in the study. The age of the subjects ranged from 21–45 years. The age structure of the sample in 5-year subgroups was as follows: 21–25 years (11.1%); 26–30 years (39.3%); 31–35 years (38.3%); 36–40 years (10.2%); and 41–45 years (2.6%). The most numerous group of respondents were women from cities with a population of over 100 thousand (45.4%). The rest of the women declared that they live in smaller towns (55.3%). Overall, 60.3% of the women were currently in a marital relationship, and for 31.7%, it was a partnership (informal); 8.0% of the respondents were not currently in any relationship. The per capita income in the household was estimated for all respondents at the current level of the minimum wage in Poland (EUR 664). Thus, the sample represented 83.2% of a sufficient standard of living. Women were asked about their current psychophysical status. None of the subjects who entered the final study pool showed a condition that would realistically affect the final outcome of the study. While 12.2% of the women were on a reduction diet prior to pregnancy, the reported BMI in their case was a normative value. Furthermore, 8.4% of the women reported having received psychological help in the past, but this help was unrelated to dietary problems. 

### 3.2. Aatropometric Data

Detailed metric and anthropometric data are presented in Table 2.

### 3.3. Factors Associated with Pregnancy and Childbirth

The first part of the questionnaire dealt with issues directly related to childbirth and how to feed the child. More than half of the respondents declared that they had given birth between 2 and a maximum of 6 months before participating in the study (*n* = 191; 66.2%). For the majority of women surveyed, this was their first birth (*n* = 195; 68.3%). The most popular method of feeding in the sample was natural feeding, and this concerned 61.4% of the respondents (*n* = 177) (Table 3).

According to the guidelines taking into account anthropometric data such as body weight and height, body mass index (BMI) was calculated and categorized according to levels: underweight (*n* = 21; 7.3%), normal (*n* = 192; 66.7%), overweight (*n* = 53; 18.4%), and obese (*n* = 22; 7.6%). The majority of subjects had normal body weight. A higher percentage of overweight and obese subjects was found in women in the age range 30–35 years (*n* = 22; 41.5% vs. *n* = 10; 45.5%). On the other hand, in the age group below 30 years, the highest percentage of women with normal weight was observed (*n* = 77; 40.1%). Table 4 shows the number of study participants (*n*; %) according to body mass index. 

The respondents did not notice any changes in their diet about their reactions to psychophysical factors (*n* = 108; 37.5%). Every third woman perceived increased snacking between meals (*n* = 87; 30.2%) and in every fourth woman a lack of appetite in response to situations perceived as stressful (*n* = 70; 24.3%). However, by far, the smallest part of the sample reported overeating as a result of strong emotions (*n* = 23; 8.0%). Next, among the most common reasons causing tension, the respondents cited maternity responsibilities (*n* = 146; 50.7%) and work or home responsibilities (*n* = 87; 30.2%). To a lesser extent, stress was caused by family conflicts (*n* = 76; 26.4%), illness (*n* = 54; 18.6%), or financial conditions (*n* = 45; 15.6%).

It was found that 198 women (68.8%) were dissatisfied with their current body weight and body shape. This group consists mainly of women who gave birth by cesarean section (*n* = 150; 52.1%).

In the comparison of satisfaction with current body weight versus BMI (*n* = 138; 49.8%), women with higher BMI more often reported dissatisfaction with their body shape (H = 13.042; *p* = 0.001).

### 3.4. Relationships with Food and Selected Birth and Lifestyle Factors in Young Mothers

Using the relationship with food questionnaire, TFEQ-13, eating behaviors were examined, distinguishing food restriction, lack of control over eating, and emotional eating. These behaviors were examined in terms of factors such as satisfaction with current weight, BMI, type of delivery, and how the child was fed.

A significant relationship was found between the tendency to restrict eating and BMI. Respondents who were overweight or of normative body weight showed the greatest tendency towards dietary restraint (H = 12.238; *p* = 0.001). This was particularly evident in women who were dissatisfied with their body weight (H = 10.471; *p* = 0.007) (Table 5).

For lack of control over eating, subjects who were dissatisfied with their weight showed significantly higher severity of this factor (H = 10.573; *p* = 0.001). There was no association between this psycho-marker of weight loss and the other factors studied (Table 6).

Although no significant differences were shown, the severity of the psycho-marker of emotional eating increased in direct proportion to body mass index. This score was also higher in those who were dissatisfied with their body weight (H = 9.893; *p* = 0.001) (Table 7).

## 4. Discussion

Body image is a multidimensional construct in which various thoughts, beliefs, emotions, and behaviors play a dynamic role in one’s subjective assessment of one’s physical appearance and overall attitude toward one’s body [5]. Understanding the mechanisms by which women can achieve greater satisfaction with their bodies during pregnancy is an important research topic that can inspire planning for more effective psychological support [6].

Although abnormal weight in pregnancy, as in non-pregnant women, is defined by the BMI found before the first pregnancy visit or in the first trimester, this index will not always be an ideal indicator of weight gain for a pregnant woman and actual fat mass. It would probably be much more useful to assess weight gain by trimester. However, since the study was based on women’s declarations, this type of question was abandoned and could not be confirmed with patients’ medical records. However, it is undisputed that weight gain during pregnancy should depend on a woman’s baseline weight before pregnancy. Recommendations from the Institute of Medicine and the American Society of Obstetricians and Gynecologists show normal weight gain in pregnant women based on BMI and pre-pregnancy weight. For women who are underweight before pregnancy (BMI < 18.5), the indicated normal weight gain is 12.7 kg to 18.2 kg during pregnancy. For normal weight (BMI = 18.5–24.99), the gain should be 11.4–15.9 kg, while for a group of women who are overweight (BMI = 25.0–29.99), the indicated weight gain during pregnancy is 6.8–11.4 kg. For obese women with a BMI > 30 kg/m^2^, weight gain during pregnancy should not be more than 7 kg, but for pregnant women who have a BMI > 40 kg/m^2^, weight reduction should be recommended. In women with prevalent obesity, better obstetric outcomes have been shown when weight was reduced by either 0.19 kg per week during pregnancy or 7.6 kg over the course of the entire pregnancy. The results of scientific studies show that these norms may not be applicable to overweight pregnant women and those with obesity and coexisting gestational diabetes. Some studies show that a smaller weight gain is safe and may have a beneficial effect on obstetric outcomes in women with an elevated BMI; a weight gain of 0–2.2 kg was associated with a lower rate of macrosomia [28,29,30,31,32,33,34].

In the results described above, it was observed that focus on food restriction and lack of control over eating as well as emotional eating in the sample of women were related to negative body image perception in the form of weight dissatisfaction. Relationships with eating showed an association with BMI level and weight satisfaction after pregnancy. This study addresses potential eating disorders in the context of eating relationships after pregnancy ending in childbirth by pointing to changes in physical body image. This includes an increased anthropometric score of approximately 3.5 kg of postpartum weight gain, at the same time indicating an increased percentage of women with a body mass index of more than 25 kg/m^2^ characterized unequivocally by dissatisfaction with their own body (68.8%). Similarly, studies by other authors highlight changes in women’s body image associated with increased BMI as a factor of restrictive, non-physiological dietary changes aimed at excessive weight reduction after pregnancy instead of implementing a balanced dietary model with reduced stressors before and during pregnancy. Body mass index is a predictor for the patho-mechanism of eating relationship disorders translating into perceived body image in extreme clinical cases of dysmorphophobia among postpartum women [35,36,37].

Proper nutrition during pregnancy is a very important factor for the health of mother and baby. Women during pregnancy and the postpartum period should take great care to maintain good eating habits. Proper nutrition benefits the development and health of the baby as well as the health of the mother. A proper diet during this period should contain adequate amounts of protein, carbohydrates, healthy fats, and vitamins [38]. Studies have shown an association between nutrient intake during pregnancy and the behavior and emotional functioning of the offspring [39]. A study conducted by Bojar et al. on pregnant women showed that their diet was average. Moreover, it was noted that the younger the woman (up to 25 years of age), the worse were her evaluated eating habits [40]. In our study, it was observed that many women had incorrect dietary habits such as irregularity of meals (75.3%), insufficient fluid intake (no more than 1.5 liters in 43.8% of women), and frequent snacking between meals (82.3%). It is recognized that the importance of perinatal nutrition has a great impact on the health of the offspring [41,42].The perinatal period represents a special time in the phase of psychological and physiological development of a woman despite the proven action of neuroplasticity of neurons in brain structures, which increases adaptation to hormonal, cognitive, and emotional changes. All this affects disturbances of the daily rhythm (insomnia), anxiety, and progressive weight gain, resulting in lowered self-esteem and lack of self-acceptance: dysmorphophobia [43,44,45,46]. These factors may signal the progression of new mental dysfunctions and the intensification and recurrence of psychiatric–psychological comorbidities (bipolar, anxiety, depressive disorders). The mechanism of this dysfunctional process consequently creates a risk for the development of eating disorders in women and, as a result of metabolic programming, intensifies the patho-mechanism of neuropsychiatric polyplatologies in the offspring, e.g., neurotransmission disorders and psychomotor hyperactivity. The immuno-psycho-protective role is also played by the method of feeding the newborn using the natural method and the type of delivery in the form of natural forces (natural childbirth). However, in the conducted self-analysis, the significance of significant for the three types of behaviors in the context of eating disorders was not obtained [45,47,48,49,50]. Scientific reports confirm that diet unequivocally affects the gut–brain axis, contributing to the health-promoting regulation of women’s mental health during the prenatal and postnatal periods. Data indicate that approximately 25.0% of psychopathology affects women during the perinatal period, increasing predisposition to the patho-mechanism of nutritional dysfunction and negative clinical picture of the pre-pregnancy period and remaining latent during this period. On the other hand, in the postpartum phase, it intensifies the progression of previously diagnosed eating disorders, manifesting itself, for example, in the form of vomiting with a frequency of about 30.0% [51,52,53,54,55,56]. Eating disorders in the form of anorexia nervosa (anorexia nervosa), bulimia (negative caloric balance), and targeted overweight and obesity (positive caloric balance) exacerbate destructive changes in women’s behavior. In addition, including remission of anorexia nervosa and bulimia nervosa during pregnancy due to a lower diagnosis rate of 5.0% compared to the postpartum period of 15.0% postpartum, especially at both six weeks and six months [35,56,57]. The TFEQ-13 questionnaire was used in the study. The methodology of the TFEQ-13 questionnaire makes it possible to assess the relationship in the context of eating and psycho-emotional factors, taking into account three types of eating behavior: restriction towards food, lack of control over eating, and emotional eating. A cross-sectional study by authors Kowalkowska and Poínhos conducted in a general population group of young women and men show a significant correlation between body mass index (BMI) and emotional eating and food restriction with a predominance among women (*p* = 0.001). In contrast, the coefficient of the level of social attractiveness was found to be lower for uncontrolled consumption and under the influence of emotional factors with a predominance among women (*p* = 0.001). Implemented physical activity in a significant percentage of women was a significant correlation between food restriction (*p* = 0.008) as well as taking into account the BMI index (*p* = 0.005), while in the case of BMI and emotional eating, a strong correlation was also found (*p* = 0.011). Factors in the form of social acceptance, level of physical activity implemented, and BMI strongly influence eating behavior distinguishing in turn: uncontrolled eating, excessive control, or emotional consumption [58]. The final analysis of the TFEQ-13 score makes it possible to assess in the context of the perception of eating under the influence of external emotional situations, both positive and negative, in individuals with the possible development of eating disorders [58,59]. Recalling a self-analysis performed among a group of postpartum women, they were not satisfied with their body image in both restrictive eating (*p* = 0.001), uncontrolled eating (*p* = 0.068). Their relationship with food showed a significant relationship with BMI.

Emotional eating is definitely an interesting and ambiguous aspect in the studied population of women because a minimal effect on body mass index was observed in the case of emotionally influenced eating, together with deprivation of immune-protective properties (artificial feeding, delivery by caesarean section) in the bodies of women in the perinatal period. In addition, a statistically significant relationship between pejorative perception of body image (*p* = 0.001) and loss of control over eating under the influence of emotional state. Therefore, further in-depth analysis of the psychodietetic aspect in the context of loss of control over food consumption after distress and eustress is required.

## 5. Strengths and Limitations

Undoubtedly, a strength of the conducted study is the large group of recipients, which consisted of a total of 288 mothers, which constituted a representative group as far as studies conducted in these groups are concerned. Using appropriately selected methods, it was possible to collect a group homogeneous in terms of the time of the last birth, which greatly reduced the researcher’s error.

The study used the CAWI method, which in some circles may be considered unreliable due to the lack of control of the researcher over the course of the survey; however, in the case of the tool used, we have a high degree of confidence that the questionnaires were filled out reliably, which was also confirmed during the pilot study and validation of the survey.

Unfortunately, the study was very unidimensional, so in further projects, it is planned to test the level of postpartum depression in a group of young mothers and correlate it with the occurrence of pathological eating behaviors. It is also noted that the method of examining satisfaction with one’s own body (body weight) by asking questionnaire questions may be unreliable, and more detailed methods (face-to-face interview, methods involving comparison of figure to illustrations, anthropometric methods, etc.) should be used in the future. Studies emphasize that questionnaire-based assessment of body satisfaction may be inadequate because most people underestimate or misestimate their body size [60].

The use of BMI to categorize anthropometric data is also questionable, as there is a high probability of losing some information. Nevertheless, categorization by BMI was used to improve the reader’s understanding of the data. The use of raw data (weight and height) without a proper description and a matching descriptive category could be unreadable.

Further research is planned to compare the study characteristics in terms of sociodemographic data such as family status (married, civil union, single), level of welfare receipt, and economic data on income.

## 6. Conclusions

Based on the study, it was found that the focus on food restriction and lack of control over eating as well as emotional eating in the study sample of women had some association with negative perceptions of body image in the form of weight dissatisfaction. Eating behaviors showed an association with BMI and weight satisfaction after pregnancy. Eating behaviors showed an association with BMI level and post-pregnancy weight satisfaction. A tendency toward snacking was observed in the overall eating characteristics of the women studied although this aspect would require confirmation in further studies. However, it is worth noting that the most common cause of stress reactions is maternal duties, which predispose to excessive eating. The conducted study shows the relevance of considering the aspect of altered perception of body image by women after childbirth predisposing to eating disorders in the form of emotional eating, lack of control, and restriction of food intake. The perception of body image by postpartum women in the context of the eating habits of women after childbirth is important, as the emotional state affects the nutritional status of the study population.

## Figures and Tables

**Table 1 ijerph-19-10137-t001:** Interpretation of scores for psycho-markers according to TFEQ-13.

Psycho-Marker	Subscale Questions
Restricting food	1, 9, 10, 12, 13
Lack of control over eating	2, 5, 6, 7, 11
Eating under the influence of emotions	3, 4, 8

**Table 2 ijerph-19-10137-t002:** Data on age and anthropometric measures in the sample (*n* = 288).

Variable	X	SD	Min.	Max.
**Age**	29.6	4.3	20.0	45.0
**Bodyweight before pregnancy (kg)**	61.3	10.2	39.0	110.0
**Bodyweight after pregnancy (kg)**	64.8	12.4	42.0	122.0
**Height (cm)**	166.9	5.87	150.0	182.0

X, average; SD, standard deviation; Min.—minimum value; Max., maximum value.

**Table 3 ijerph-19-10137-t003:** Characteristics of the group regarding the date of delivery, number of pregnancies, and feeding method used (*n* = 288).

The Distribution Criterion	Categories	*n*	%
**Date of last delivery**	Up to 2 months ago	32	10.8
	2–6 months ago	191	66.2
	7–12 months ago	65	23.0
**Type of delivery**	Natural	130	44.5
	Cesarean section	158	55.5
**Number of children born**	First	195	68.3
	Next	93	31.7
**Method of feeding**	Artificial mixture	61	21.1
	Partial breastfeeding	50	17.5
	Breastfeeding	177	61.4

**Table 4 ijerph-19-10137-t004:** Number of subjects according to BMI by WHO (*n* = 288).

BMI	*n*	%
**Underweight (<18.5 km/m^2^)**	21	7.3
**Normal weight (18.5–24.9 kg/m^2^)**	192	66.7
**Overweight (25.0–29.9 kg/m^2^)**	53	18.4
**Obesity (>30.0 km/m^2^)**	22	7.6

**Table 5 ijerph-19-10137-t005:** Food restriction and selected variables related to birth and current maternal weight (*n* = 288).

Variable	*n*	%	M	Min.	Max.	QR	*p*-Value
Food restriction vs. **BMI**							
Underweight	21	7.3	2.0	0	11	3.0	*p* = 0.001 *
Standard	192	66.7	5.0	0	14	5.0	
Overweight	53	18.4	6.0	0	12	5.0	
Obesity	22	7.6	4.0	1.0	11	5.0	
Food restriction vs. **Satisfaction with current weight**							
Not	198	68.8	5.0	0	14	5.0	*p* = 0.007 *
Yes	90	31.2	4.0	0	11	4.0	
Food restriction vs. **Type of delivery**							
Natural	158	44.5	5.0	0	14	4.0	*p* = 0.296 *
Cesarean section	130	55.5	5.0	0	13	5.0	
Food restriction vs. **Method of feeding**							
Breastfeeding	177	21.1	5.0	0	14	5.0	*p* = 0.721 *
Partial breastfeeding	50	17.5	5.0	0	12	5.0	
Artificial mixture	61	61.4	5.0	0	12	4.0	

M, median; Min., minimum value; Max., maximum value; QR, quartile range. * *p*-value for Kruskal–Wallis test.

**Table 6 ijerph-19-10137-t006:** Lack of control over eating and selected variables related to birth and current maternal weight (*n* = 288).

Variable	*n*	%	M	Min.	Max.	QR	*p*-Value
Lack of control over eating vs. **BMI**							
Underweight	21	7.3	7.0	0	11	4.0	*p* = 0.068 *
Standard	192	66.7	6.0	0	15	4.0	
Overweight	53	18.4	6.0	0	13	4.0	
Obesity	22	7.6	8.0	3.0	15	4.0	
Lack of control over eating vs. **Satisfaction with current weight**							
Not	198	68.8	7.0	0	15	5.0	*p* = 0.001 *
Yes	90	31.2	5.0	0	15	3.0	
Lack of control over eating vs. **Type of delivery**							
Natural	158	44.5	6.0	0	15	4.0	*p* = 0.741 *
Cesarean section	130	55.5	6.0	0	14	4.0	
Lack of control over eating vs. **Method of feeding**							
Breastfeeding	177	21.1	6.0	0	15	4.0	*p* = 0.512 *
Partial breastfeeding	50	17.5	6.5	0	14	3.0	
Artificial mixture	61	61.4	6.0	0	15	5.0	

M, median; Min., minimum value; Max., maximum value; QR, quartile range. * *p*-value for Kruskal–Wallis test.

**Table 7 ijerph-19-10137-t007:** Eating under the influence of emotions and selected variables related to birth and current maternal weight (*n* = 288).

Variable	*n*	%	M	Min.	Max.	QR	*p*-Value
Eating under the influence of emotions vs. **BMI**							
Underweight	21	7.3	3.0	0	7	4.0	*p* = 0.094 *
Standard	192	66.7	3.0	0	9	4.0	
Overweight	53	18.4	4.0	0	9	4.0	
Obesity	15	7.6	4.0	1.0	8	4.0	
Eating under the influence of emotions vs. **Satisfaction with current weight**							
Not	198	68.8	3.0	0	9	4.0	*p* = 0.001 *
Yes	90	31.2	2.5	0	9	4.0	
Eating under the influence of emotions vs. **Type of delivery**							
Natural	158	44.5	3.0	0	9	3.0	*p* = 0.895 *
Caesarean section	130	55.5	3.0	0	9	4.0	
Eating under the influence of emotions vs. **Method of feeding**							
Breastfeeding	177	21.1	3.0	0	9	4.0	*p* = 0.332 *
Partial breastfeeding	50	17.5	3.5	0	9	3.0	
Artificial mixture	61	61.4	3.0	0	9	4.0	

M, median; Min., minimum value; Max., maximum value; QR, quartile range. * *p*-value for Kruskal–Wallis test.

## Data Availability

Not applicable.

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
