# Peer review of "Perception of the Body Image in Women after Childbirth and the Specific Determinants of Their Eating Behavior: Cross-Sectional Study (Silesia, Poland)"

_ijerph, 2022, doi:10.3390/ijerph191610137_

Round 1

Reviewer 1 Report

The review results for this study are as follows:

< Background>

1.     In the study setting, BMI levels for pregnant or postpartum women are generally excluded. In this study, the basis for applying BMI level as an independent variable should be described.

2.     Everyone understands and accepts that pregnancy or postpartum period involves a change in body shape. Changes in body image caused by pregnancy are taken for granted as a normal response. However, the authors argue that “pregnancy and the postpartum period may be an additional determinant of eating disorder”. Describe what this basis is.

3.     In order to develop this study, the authors must provide scientific evidence for the reasoning that pregnant or postpartum women will be induced with eating disorders due to their body image.

<Materials and Methods>

4.     Describe the recruitment process for the study participants.

5.     The timing of the survey of pregnant or postpartum women, i.e. weeks of pregnancy, weeks or months after childbirth, should be described. This is because the weeks or postpartum period of pregnancy can affect the body shape or body image. The timing of women's body shape returning to normal after delivery should be considered.

< Discussion & Conclusions>

6.     Eating behaviors showed an association with BMI level and weight satisfaction after pregnancy. This conclusion is thought to be possible when the authors prove that the basis for applying BMI levels to pregnant or postpartum women is valid.

Author Response

Dear Reviewer,

Thank you for reviewing our article and for your many valuable and substantive comments.

The answers are marked in blue in the text on the following lines:

  1. 329-351

2-3. 80-110

4-5. 175

  1. We hope that the answers given, will allow you to consider the proposals as legitimate.

We hope that the changes made will allow our article to be accepted for publication.

Thank you again for your help.

With best regards, Authors

Reviewer 2 Report

The authors performed a study about the perception of body image by postpartum women and the specific determinants of their eating behaviour.

1. The title should be simplified as it is confusing as it is. The authors should state what kind of study they performed in the title.

2. In the introduction the authors are kindly requested to state the purposes of this study, replacing the questions posed in the last paragraph.

3. The authors are kindly requested to acknowledge the type of study they conducted and to follow the appropriate EQUATOR Statement.

4. The number of registration of the Ethics Committee should be mentioned as they received it.

5. The methods section needs to be thoroughly revised and rewritten. Numbers do not belong to the methods. Numbers and report of results belong to the results.

6. What does this study add to the current knowledge? The authors should state this fact in the Conclusions section.

Author Response

Dear Reviewer,

Thank you for reviewing our article and for your many valuable and substantive comments.

The answers are marked in red in the text on the following lines:

  1. 1-4
  2. 130-139
  3. 147-234
  4. 175
  5. 235-256
  6. 446-480

We hope that the changes made will allow our article to be accepted for publication.

Thank you again for your help.

With best regards, Authors